# In Vitro Binding Effects of the Ecdysone Receptor−Binding Domain and PonA in *Plutella xylostella*

**DOI:** 10.3390/molecules28031426

**Published:** 2023-02-02

**Authors:** Yanjiao Feng, Jialin Cui, Binyan Jin, Xiuzhen Li, Xiaoming Zhang, Libing Liu, Li Zhang

**Affiliations:** 1Innovation Center of Pesticide Research, Department of Applied Chemistry, College of Science, China Agricultural University, Beijing 100193, China; 2Department of Nutrition and Health, China Agricultural University, Beijing 100193, China

**Keywords:** ecdysone receptor, ecdysone agonist, binding effect, *Plutella xylostella*, molecular modeling

## Abstract

Both insect ecdysone receptors and ultraspiracle belong to the nuclear receptor family. They form a nanoscale self-assembling complex with ecdysteroids in cells, transit into the nucleus, bind with genes to initiate transcription, and perform specific biological functions to regulate the molting, metamorphosis, and growth processes of insects. Therefore, this complex is an important target for the development of eco-friendly insecticides. The diamondback moth (*Plutella xylostella*) is a devastating pest of cruciferous vegetable crops, wreaking havoc worldwide and causing severe economic losses, and this pest has developed resistance to most chemical insecticides. In this study, highly pure EcR and USP functional domains were obtained by constructing a prokaryotic expression system for the diamondback moth EcR and USP functional domain genes, and the differences between EcR and USP binding domain monomers and dimers were analyzed using transmission electron microscopy and zeta potential. Radioisotope experiments further confirmed that the binding affinity of PonA to the EcR/USP dimer was enhanced approximately 20-fold compared with the binding affinity to the *Px*GST−EcR monomer. The differences between PonA and tebufenozide in binding with EcR/USP were examined. Molecular simulations showed that the hydrogen bonding network formed by Glu307 and Arg382 on the EcR/USP dimer was a key factor in the affinity enhancement. This study provides a rapid and sensitive method for screening ecdysone agonists for ecdysone receptor studies in vitro.

## 1. Introduction

Nuclear receptors are a class of ligand-dependent superfamilies of transcriptional regulators located in the nucleus. They regulate the promoter-, ligand- or cell type-specific expression of different target genes and participate in a variety of physiological processes, such as embryonic development, cell differentiation, intracellular environmental stability and energy homeostasis [1,2]. The activity of the nuclear receptor can be regulated by the binding of the corresponding ligand. The ligand is a small lipophilic molecule that readily penetrates biological membranes and forms a nanoscale self-assembling complex with the nuclear receptor protein upon entry into the cell, which binds to the gene promoter region after translocation into the nucleus and initiates transcription to perform its specific biological function [3]. Therefore, nuclear receptors are considered excellent drug targets. In insects, the ecdysone receptor that regulates the metamorphosis process is a nuclear receptor. The commercialized insecticide tebufenozide kills some lepidopteran and coleopteran pests by interfering with the normal molting process [4].

The diamondback moth, *P. xylostella* (L.) (Lepidoptera: Plutellidae) has become one of the most destructive vegetable pests in the world due to its severe damage to cruciferous plants, resulting in an annual economic loss of approximately 4–5 billion per year in agricultural production [5]. Currently, chemical control is the main measure to manage the diamondback moth [6]. However, the heavy use of insecticides has accelerated the development of resistance in *P. xylostella* (L.) [7]. Therefore, there is an urgent need to develop new insecticides with high efficiency and a low risk of resistance. The ecdysone analog PonA, which targets the insect-specific molting process, has received extensive attention. As an important example of insect growth regulators, PonA interacts with EcR/USP, a heterodimer, and initiates downstream transcriptional elements of ecdysteroids with the assistance of transcriptional activators, triggering an internal regulatory cascade that triggers molting and metamorphosis processes in insects [8,9]. However, the complex structure of PonA and its difficult synthesis have limited its further application. Ecdysone agonists, including commercialized dibenzoylhydrazine insecticides, have been developed to mimic the activity of ecdysteroids and interact with ecdysone receptor proteins [10], resulting in the continuous excessive molting and death of insects [11]. In recent years, the extensive cloning and characterization of EcR and USP gene sequences in different insects have provided the foundation to study receptor functions and ecdysone agonist screening [12,13,14].

EcR has the structural features of a typical nuclear receptor, consisting of five parts: an A/B structural domain (transcriptional activation domain), a C structural domain (DNA binding domain), a D structural domain (hinge region), an E structural domain (ligand binding domain), and an F structural domain [15]. Among them, the C domain is the most conserved and can recognize specific target gene sequences; the D domain forms dimers between EcR and USP, which contains nuclear localization signals in many nuclear receptors; and the E domain is the second most conserved, formed by 12 α-helices and β-folds and is a binding pocket that accommodates ligands such as ecdysone. When ligands enter this E domain, they interact with amino acid residues inside the pocket, leading to a change in its spatial conformation and thus activating gene transcription [16]. The strength of the interaction between different ligands and pocket formation is closely related to the type of amino acid residue [17]. USP, which is homologous to EcR, has the same structural features as EcR except for the absence of the F domain [18]. It has been shown that the affinity of heterodimers formed by expressing the LBD functional structural domains of EcR and USP are the same as those of heterodimers formed by the proteins of EcR and USP expressed from the full-length genes; the binding of heterodimers to ligands can thus be determined using partial gene fragments to express the amino acid sequence of the ligand-binding pocket [19,20]. To date, in vitro activity testing methods for ecdysteroids and their analogs have included radioligand binding and emerging reporter gene assays [21,22,23], of which radioligand binding remains the “gold standard” and an indispensable tool for characterizing the structure-activity relationship of new test compounds [24]. The in vitro transcription-translation protein system combined with the radioligand assay was also used in our laboratory to test the in vitro activity of the commercial insecticide tebufenozide and ecdysone agonists with EcR [25]. However, the purity of the target proteins obtained by this method was quite low, and there were many heterogeneous proteins [26], which interfered with the sensitivity and accuracy of the in vitro activity test at the protein level.

In the present study, we obtained highly pure functional structural domain proteins of the ecdysone receptor *Px*EcR/USP of *P. xylostella* (L.) by constructing in vitro prokaryotic expression vectors for the functional domain genes of EcR and USP and characterized their monomeric and dimeric properties using transmission electron microscopy and zeta potential. In addition, the differences in their binding to different ligands were investigated by radioisotope experiments and theoretically explained using molecular simulations (Figure 1). This study provides a sensitive and efficient method for testing the binding activity of ecdysone agonists to EcR.

## 2. Results and Discussion

### 2.1. Identification of the Target Proteins PxGST-EcR and PxGST-USP for Prokaryotic Expression

The diamondback moth ecdysone receptor protein (*Px*EcR/USP) was expressed in *Escherichia coli* as a truncated LBD-containing functional domain fused to glutathione-S-transferase (*Px*GST-EcR/*Px*GST-USP), and after IPTG induction and affinity purification, the target protein was identified by SDS-PAGE and Coomassie staining on a 10% gel. The specific protein bands appeared at approximately 72 kD (*Px*GST-EcR) and 68 kD (*Px*GST-USP) (Figure 1A), and the target bands were clear in accordance with the requirements of the subsequent experiments. In addition, the results of Western blot detection of the target protein with the specific GST-tagged antibody are shown in Figure 1B, and the molecular weight where bands appeared matches the theoretical molecular weight of the target protein. Therefore, both the molecular weight and specific antibody detection results of the purified proteins indicate that they are the target proteins. The purity of the target protein obtained by affinity purification was greatly enhanced in this study compared with the previously reported results of in vitro transcription–translation of the target protein [26]. This in turn reduced the interference factors in the subsequent experiments.

The final yields of the purified target proteins were 590 µg·L^−1^ culture (*Px*GST−EcR) and 2.7 mg·L^−1^ culture (*Px*GST−USP), indicating that the expression of *Px*GST−EcR was lower than that of *Px*GST−USP, which is consistent with the results of Grebe M et al., who used *Chironomus tentans* to express *Ct*EcR/USP and purified the resulting proteins; however, the expression levels of both target proteins were higher than those in *C. tentans* [27]. In addition, Rusin et al. enhanced the solubility of the receptor structural domain with six histidine tags in *Drosophila melanogaster*, with elevated expression of 11% (*Dm*EcR−BD) and 16% (*Dm*USP−BD) [28], demonstrating the effect of different tags and different expression regions on the expression of the target protein. In our subsequent experiments, we also found that the expression and purity of both target proteins were further enhanced, particularly the expression purity, when the flexible partial sequences in front of the functional structural domains of *P. xylostella* (L.) EcR and USP were removed for expression.

### 2.2. TEM Observation of Target Protein Monomer and Dimer

Previous studies have reported that both crystal structure analysis of the ecdysone receptor EcR/USP and mass spectrometry results indicate that the ecdysone receptor exerts its physiological function as a heterodimer [29]. To demonstrate whether the two target proteins (*Px*GST−EcR and *Px*GST−USP) expressed in vitro in prokaryotic form can polymerize to form heterodimers to exert their functions, we used TEM to separately observe the morphology of the target protein monomer and its mixed sample after incubation; this technique uses the dispersion of electrons deflected by the material to produce grayscale images, allowing observation of the precise morphology of the target protein at higher magnification and clarity than other types of microscopes. The images obtained are shown in Figure 2, from which it can be observed that both monomeric proteins *Px*GST−EcR (Figure 2A) and *Px*GST−USP (Figure 2B) show irregular to almost spherical structures with sizes ranging from approximately 10 nm to 20 nm, whereas the dimeric protein samples (Figure 2C) after incubation with both monomers also show irregular shapes with protein sizes ranging from approximately 20 nm to 40 nm. From the image, we can see that the monomer *Px*GST−EcR is larger than *Px*GST−USP, which is also consistent with the molecular weight results of the two target proteins. Furthermore, the monomeric proteins were also subjected to incubation experiments under the same conditions; however, autodimerization was not observed for either of the two monomeric proteins. Thus, GST tagging was excluded as a possible cause of dimerization in some proteins, and the size of the dimeric proteins in this image was found to be approximately twice that of the monomeric proteins, reflecting the formation of heterodimers from the two monomeric proteins after 90 min of incubation at 25 ± 1 °C in vitro.

### 2.3. Zeta Potential Detection

In this study, coincubation samples of the target protein monomers of *Px*GST−EcR and *Px*GST−USP and incubation of samples of the individual monomers for zeta potential tests were performed at room temperature and further illustrated the formation of *Px*GST−EcR/USP heterodimers. The results of the zeta potential assay are shown in Table 1: the potentials of monomeric proteins *Px*GST−EcR and *Px*GST−USP were −4.52 ± 0.56 mV and −4.46 ± 0.01 mV, respectively; with the formation of heterodimers by coincubation of monomeric proteins, there was a significant negative shift in the potential of the samples, which was more negative than the zeta potential of each individual monomer. The zeta potential of the dimer was −8.56 ± 1.7 mV, thus indicating the formation of a heterodimer from the two monomeric proteins under the coincubation conditions and resulting in a significant change in the electrical properties of the solution. These data are consistent with the results of the TEM experiments, demonstrating the dimerization of the two monomeric proteins during incubation. It was also found that when the natural ligand PonA was added to the heterodimer solution at 10 nmol·L^−1^, the overall solution potential was altered, and the potential shifted positively due to amino acid interactions between the ligand and the dimer ligand binding domain, which further changed the original forces of the amino acids around the empty dimer ligand binding domain.

### 2.4. Specific Binding Experiment of [^3^H]PonA with EcR/USP

To further evaluate the biological activities and binding patterns of the target proteins (*Px*GST−EcR and *Px*GST−USP) after exogenous expression purification, we tested the binding of the two target proteins to the natural ecdysteroid [^3^H]PonA in different combinations, as shown in Figure 3A. The experimental results showed that [^3^H]PonA did not bind to the monomeric protein *Px*GST−USP (i.e., there was no significant difference between total and nonspecific binding) but bound specifically to *Px*GST−EcR; in addition, the binding of *Px*GST−EcR to [^3^H]PonA was significantly enhanced in the presence of *Px*GST−USP (*Px*GST−EcR and *Px*GST−EcR/USP dimer-specific binding of 345 dpm and 1179 dpm, respectively) by approximately four-fold (Figure 3A), which was consistent with previous tests [21,30].

It has been reported that after separate expression of the ligand−binding structural domains of *Dm*EcR and *Dm*USP, recombination for binding to the ligand is inactive [31]: only coexpression of the two proteins can produce a correctly folded dimeric protein structure for binding to the ligand, and this dimer cannot be recovered by subsequent mixing of the two. However, our previous study showed that separate expression of the functional structural domains of *Px*EcR and *Px*USP followed by remixed dimerization resulted in a hormonally active dimer that can bind its ligand [30]. In the present study, our experimental data further demonstrate that [^3^H]PonA can bind to either *Px*GST−EcR or *Px*GST−EcR/USP dimers. In further saturation binding experiments, the dissociation equilibrium constant K_d_ value of 48.8 nM and B_max_ of 1433 dpm for [^3^H]PonA with monomeric protein *Px*GST−EcR was calculated using specific binding saturation curves (Figure 3B), whereas the dissociation equilibrium constant K_d_ value of 2.3 nM and B_max_ of 1497 dpm were calculated for the *Px*GST−EcR/USP dimer (Figure 3C). The binding affinity of PonA to *Px*GST−EcR in the dimer was enhanced approximately 20-fold compared with the affinity to *Px*GST−EcR in the monomer, indicating that *Px*GST−USP enhances the binding of the ligand to *Px*GST−EcR. This result is in general agreement with previously reported results [21,26]. In addition to *P. xylostella* (L.), Morishita, C. M. et al. reported that PonA could bind to the EcR/USP complex from *Harmonia axyridis* and *Epilachna vigintioctopunctata*, but tebufenozide was inactive against *E. vigintioctopunctata* [32]. These experiments further showed that the binding activity of PonA to EcR/USP complexes of different species was greater than that of tebfenozide, which bound only partially to EcR/USP complexes in lepidopteran and coleopteran insects.

### 2.5. Binding of Tebufenozide to EcR/USP

After testing the activity of the purified target protein and the binding activity of the natural ecdysteroid PonA, we also tested the in vitro activity of a typical ecdysteroid analog control insecticide, tebufenozide, using the radioligand binding method by incubating the reaction with different concentrations of tebufenozide at a target protein concentration of 300 µg·mL^−1^ and [^3^H]PonA 10 nmol·L^−1^ and performing a competitive binding assay. As the concentration of tebufenozide increased, its ability to inhibit [^3^H]PonA increased such that the detectable [^3^H]PonA bound to *Px*GST−EcR/USP decreased; the results are shown in Figure 3D. The IC_50_ value of tebufenozide for this heterodimer was 80.58 nmol·L^−1^, which is approximately ten-fold lower than our previous results with transcriptional translation of the target protein (IC_50_ = 0.85 µM) [25], indicating that the sensitivity of our experimental method for target in vitro activity detection showed a significant improvement over that in our previous work. The high sensitivity facilitates the selection of relatively less active but more structurally novel potential compounds in the subsequent screening of candidates, which are often reported to be micromolar concentrations [33,34]. Another advantage of this method is that the interference of signal transduction pathway components downstream of the EcR/USP complex in binding experiments will not be considered. Some previous reports showed that the bell-shaped concentration response curves observed in ligand screening experiments performed in insect cell lines and cultured insect tissues are mainly due to the presence of the intrinsic 20E signaling pathway [35,36]. Our experiments revealed a single concentration response curve, which is more conducive to precise analysis of ligand—receptor interactions.

### 2.6. The Binding Mechanism of PxEcR with Different Ligands

We found that the binding of *Px*EcR/USP to the natural molting hormone PonA was significantly stronger than the binding to tebufenozide, based on the results of the above in vitro tests of the binding of *P. xylostella* (L.) EcR/USP and two different ligands (the natural molting hormone PonA and the commercial insecticide tebufenozide). To understand the reasons for the differences in their binding, we carried out a molecular simulation to analyze the binding mode and mechanism of the two ligands. We analyzed the binding mode and mechanism of PonA in *Px*EcR−LBD from the perspective of molecular modeling and obtained the binding conformation of PonA in *Px*EcR−LBD using molecular docking and structural clustering. Conformational superposition revealed that tebufenozide and PonA bound in two different active pockets of *Px*EcR−LBD (Figure 4A), and PonA showed a higher binding affinity for *Px*EcR−LBD (Figure 4B). This is consistent with the experimental results of this study in which tebufenozide and PonA were measured.

By further analyzing the mutual mechanism of the two pockets, we found that there is an overlap between these two pockets, and Asn503 in this region is also able to form a common H−bond with a distance of 2.96 Å to both ligands (Figure 4A). This suggests that Asn503 is an important amino acid for the binding of *Px*EcR to both ligands. In addition, tebufenozide also exhibits hydrogen bond interactions with Tyr407 and Thr342, and the force of action is mainly concentrated on the backbone structure (Figure 4C). PonA was able to form more hydrogen bonds with the surrounding Glu307, Arg382, and Ala397 due to its five hydroxyl and one carbonyl groups (Figure 4D). Notably, Glu307 and Arg382 formed a hydrogen bonding network with the two hydroxyl groups in the tail of PonA, which made the binding of PonA to the pocket more stable.

Thus, PonA is more stably bound in the active pocket of *Px*EcR through more hydrogen bonding interactions than tebufenozide. The formation of a hydrogen bonding network with Glu307 and Arg382 is the key factor. Additionally, for both binding mechanisms, the ligands were able to form hydrogen bonds with Asn503. Asparagine plays an important role in the binding of steroid and nonsteroid ligands to EcR of different species. For example, *Tribolium confusum Tc*EcR Asn521 [37], *Bemisia tabaci Bt*EcR Asn309 [38], Heliothis virescens *Hv*EcR Asn504 [29,32,39], and Harmonia *axyridis Ha*EcR Asn207 [32].

## 3. Materials and Methods

### 3.1. Chemicals

^3^H−PonA (radioactive specific activity 95 Ci·mmol^−1^, concentration 1 mCi·mL^−1^, 1.05 × 10^−5^ mol·L^−1^, PerkinElmer Inc., Shelton, CA, USA) and 95% tebufenozide (analytical pure) were used. The scintillation solution was 2,5-diphenyloxazole (PPO) and P-phenylene benzoxazole xylene.

### 3.2. Transformation and Expression Identification of Recombinant Plasmids in E. coli

Prokaryotic expression recombinant vectors pGEX-6P-1-EcR and pGEX-6P-1-USP containing optimized target genes (*Px*EcR/USP) were transformed into Rosetta receptor cells and coated on plates spiked with chloramphenicol and ampicillin, respectively (chloramphenicol 20 mg·L^−1^, ampicillin 50 mg·L^−1^) [40]. The culture was inverted overnight at 37 °C, inoculated in LB liquid medium (10 g·L^−1^ tryptone, 5 g·L^−1^ yeast extract, and 10 g·L^−1^ NaCl) on the next day for activation, and incubated at 37 °C with LB liquid medium at a ratio of 1:100 until the OD_600_ was 0.8~1.0. Isopropyl β-d-thiogalactoside (IPTG) was added at final concentrations of 0.25 mmol·L^−1^ and 0.5 mmol·L^−1^ to continue the induction culture for 6 h and 16 h and then centrifuged to collect a small amount of bacteria, which were suspended by shaking with Tris buffer (pH 8.0), and a quarter volume of 5 × SDS loading buffer was then added, denatured at 100 °C for 5 min, and identified by SDS-PAGE electrophoresis and Western blot analysis.

### 3.3. Purification of Target Proteins

After the determination of *Px*GST−EcR and *Px*GST−USP by denatured polyacrylamide gel electrophoresis, the mass culture contained the target protein expression vector Rosetta strain. At room temperature, sufficient bacteria were collected by centrifugation at 12,000 rpm. Fifty milliliters of Tris buffer (containing 0.02 mol·L^−1^ Tris and 0.15 mol·L^−1^ NaCl at pH 7.4) was used to melt and centrifuge the collected bacteria into a precipitate, and the protease inhibitor and 1 mmol·L^−1^ DTT were then added. At 4 °C, the *E. coli* bacteria were crushed by a homogenizer under high pressure 4–5 times. The bacterial solution was then centrifuged at 12,000 rpm at 4 °C for 30 min. The supernatant was filtered by a 0.22 μm Acrodisc filter and then transferred to a GST column balanced with Tris buffer (the sample flow rate was maintained at 1 mL·min^−1^). The impurity proteins were eluted with ten times the volume of Tris buffer, and finally, the eluting buffer (containing 10 mmol·L^−1^ reduced glutathione and 20 mmol·L^−1^ Tris−HCl at pH 7.4) was eluted [41]. The target proteins at the UV peak were collected and ultrafiltered to a final concentration of 1 mg·mL^−1^ for later use.

### 3.4. TEM Observation of Target Protein Monomer and Dimer

The morphology of purified GST−EcR and GST−USP monomers and dimers of *P. xylostella* (L.) was analyzed using transmission electron microscopy (JEM−1400plus, JEOL, Tokyo, Japan) with an operating voltage of 120 kV [42]. The samples were dispersed in distilled water, and the nanomaterials were negatively stained for sample preparation: (1) The carbon-plated support film copper mesh was placed on the sealing film, a drop of sample (approximately 30 µm) was placed on the support film and remained there for approximately 10 min, and the excess solution was then blotted off with filter paper for approximately 1 min; (2) the dried support film was placed onto the sealing film, a drop of uranium peroxide acetate staining solution (2% *w*/*v*) was added for 90 s, the excess stain was aspirated with a filter paper tip and dried for 3 h, and the sample was then observed under the electron microscope. Digital Micrograph software was used for processing and analysis [43].

### 3.5. Testing of the Zeta Potential of Target Protein Monomers and Dimers

Two samples, *Px*GST−EcR/USP and *Px*EcR/USP−PonA, were dissolved with distilled water in a volume of 10 mL, and a suspension of 0.25 mg·L^−1^ was prepared, sonicated for 10 min in the ice-water mixture (ultrasound for 3 s, off for 6 s, pause for 30 s), and tested on a zeta potential analyzer (Zetasizer Nano ZS90, Malvern Instruments Ltd., Malvern, UK) at 25 °C [44]. Measurements were performed in triplicate, and the calculated and reported results were also derived.

### 3.6. Binding Experiments of Ligands with EcR/USP

Based on published methods for ligand binding assays [31], 4 µL of in vitro expressed target protein (*Px*GST−EcR, *Px*GST−USP or a mixture of both for a total of 8 µL) was placed in a silicone tube with 10 nmol·L^−1 3^H−labeled PonA (tritiated PonA, 95 Ci·mmol^−1^, PerkinElmer, Inc.) and made up to a total volume of 16 µL with a low salt buffer (20 mmol·L^−1^ HEPES, 20 mmol·L^−1^ NaCl, 10% glycerol, 1 mmol·L^−1^ EDTA, and 1 mmol·L^−1^ 2-mercaptoethanol at pH 7.9, containing peptide inhibitor, protease inhibitor, leukocyte inhibitor 1 μg·mL^−1^ and bovine serum albumin 0.5 mg·mL^−1^) and incubated for 90 min at 25 °C. The absence of a heterodimeric protein control (buffer for solubilized proteins) was used as nonspecific binding. The final concentration of solvent (ethanol or dimethylsulfoxide) in the incubation solution was less than 1% and did not affect the binding of the ligand to the receptor. After incubation, the reaction mixture was transferred to ice and immediately filtered through a nitrocellulose membrane (NC45, Merck Millipore, Burlington, MA, USA) under a vacuum filtration device followed by rinsing the membrane three times with 3 mL of ice-cold washing buffer (buffer without protease inhibitor or bovine serum protein). After air drying for approximately 10 s, the membranes were transferred to a plastic sample vial, 2 mL of scintillation solution was added and mixed by shaking in an oscillator for more than 7 h at room temperature, and radioactivity was then measured using a liquid scintillation counter Hidex- 300 s (2 min/filter, Hidex Instruments, Turku, Finland). In the saturation binding experiments, heterodimeric proteins were incubated with five concentrations of [^3^H]PonA (0.625, 1.25, 2.5, 5.0, and 10 nmol·L^−1^). Equilibrium dissociation constants (Kd) and maximum binding capacity (B_max_) were calculated by nonlinear regression using SigmaPlot 14.0 (Systat Software Inc., San Jose, CA, USA).

### 3.7. Molecular Simulation of the Binding Differences of PxEcR/USP with Different Ligands

Homology modeling and molecular docking were performed using MOE 2022.02 [45] software to investigate the binding mechanism of tebufenozide and PonA to *Px*EcR. The *Px*EcR−LBD primary sequence (AEP25402.1 residues 285 to 528) was obtained from NCBI. Because the sequence similarity between *Px*EcR−LBD and *Hv*EcR−LBD was >85%, *Px*EcR−DBH and *Px*EcR−PonA were homology modeled with *Hv*EcR−BYI08346 (PDB ID:3IXP [39]) and *Hv*EcR−PonA (PDB ID:1R1K [34]), respectively, as templates for the different binding pockets. After homology modeling, *Px*EcR−LBD and *Px*EcR−LBD were bound to tebufenozide and PonA during molecular docking, respectively. The final docking conformation was selected by the best docking scoring with Generalized Born/Volume Integral (GB/VI) methodology [46]. The numbers of generated conformations and optimized conformations were set to 100, and the optimized conformations were scored and sorted using the GBVI/WSA dG method. Seventy-four docked conformations were obtained for PonA, and 91 docked conformations were obtained for tebufenozide. The top 20 docked conformations were ranked and clustered. The docking conformations of tebufenozide in Cluster 1 and PonA in Cluster 2 were determined as the main conformations, and their interactions were analyzed and discussed further.

## 4. Conclusions

Ecdysone agonists act as insect growth regulators at key stages of pest growth and development: they change the morphology and habits of pests; inhibit their growth, development, and reproduction; and eventually lead to their death. They have the advantages of high insecticidal activity, high specificity, and safety to humans and animals and are one of the key directions for the development of green pesticides. Therefore, it is necessary to provide a more sensitive screening method for the discovery of structurally novel and efficient ecdysone agonists. In this study, we successfully obtained the target protein EcR/USP in a prokaryotic expression system using a truncated gene fragment (D and E domains) of the ecdysone receptor of the diamondback moth and characterized the physical size, zeta potential, and molecular weight of the heterodimer before and after binding to the natural ligand PonA for the first time. The affinity of the heterodimeric protein to PonA and to the commercial insecticide tebufenozide was determined and was shown to be significantly enhanced compared with the results of previous in vitro transcriptional translation protein tests and other insect-derived yeast and insect cell line transcriptional activation assays, indicating the high sensitivity of the radioligand method applied to the target protein in this study. In addition, a detailed analysis of the differences in binding affinity of the ecdysone receptor to PonA and tebufenozide was performed using a molecular simulation method. In summary, we have established a rapid and sensitive screening method for ecdysone agonists based on the ecdysone receptor level in vitro using two purified proteins, *Px*GST−EcR and *Px*GST−USP, from *P. xylostella* (L.), which can then be used to detect and analyze the detailed structure-activity relationship between ecdysone agonists and the ecdysone receptor of *P. xylostella* (L.) and can be widely used to screen active compounds, providing a powerful tool for the rational development of novel ecdysone agonists for use in insects.

## Data Availability

Not applicable.

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
