# Peer review of "In Vitro Binding Effects of the Ecdysone Receptor−Binding Domain and PonA in Plutella xylostella"

_molecules, 2023, doi:10.3390/molecules28031426_

Round 1

Reviewer 1 Report

Feng et al. describe that the interaction between the insect ecdysone receptor and Ponasterone A. There are many unclear parts, and the authors should revise the manuscript.

- Introduction seems too long. Please make it smart. For example, the topics related to the medical fields in LL31-49 and in LL 56-59 are irrelevant to the topics of this manuscript. And the description in LL 63-64 are also unnecessary.

- L73, Since the structure of PoA is strictly different from that of ecdysone or 20-hydroxy ecdysone, please rewrite this part. “Ecdysteroids” are different from “ecdysones”, and PonA is the analogue of ecdysone.

- L79 “dihydrazide” -> “dibenzoylhydrazine” ?

- L81 The meaning of the description “are difficult to isolate” is unclear.

- L126 “Escherichia. coli” -> “Escherichia coli” (italic and remove a period after Escherichia.

- L128 “Komas” -> “Coomassie”?

- L132 and Figure 1B  How did the authors prepare the specific GST-tagged antibody? The Western blotting experiment demonstrates many bands other than the target proteins, even though the specific antibodies were employed. What are these bands? Did the authors further purify the target proteins by using a gel filtration chromatography before use to remove these unnecessary proteins? And can the authors confirm the presence of the heterodimer by the gel filtration chromatography? (the molecular weight of the EcR/USP dimer is 72+68=140 kDa. The retention time of the dimer peak is expected to be clearly different from that of each monomer peak.)

- Figure 2  I cannot believe that TEM can distinguish between a monomer (70 kDa) and a dimer (140 kDa), although I have ever employed TEM before. Please cite some references, in which the sizes of proteins are distinguished using the normal TEM technologies.

- L179-197, and Table 1  Please add the descriptions about the statistical analyses. Have the authors analyzed the significant difference among the data by one-way ANOVA/t-test? If yes, please add the asterisks, *, to display the significant difference (especially for L186, “a significant negative shift”). In LL192-194, the authors write that the overall solution potential was shifted positively when PonA was added to the heterodimer solution. Does this description indicate the change of “-8.56 to -8.23”? If yes, the difference between these values seems insignificant because the standard deviation of -8.56, 1.7, is too big (overlap with -8.23). Related to these, the date of “PxGST-EcR-PonA” and “PxGST-USP-PonA” (and PonA only) should be necessary to compare the data. In addition, for which was the concentration in Table 1, total proteins or each protein? If for each protein, the authors should measure the zeta values for the monomer proteins at 0.5 ug/mL. Please write the concentration of the ligand PonA employed? Please describe the influence of the 10 min sonication (L341) on the proteins. (Continuously sonicated? If yes, the solution will become hot, denaturing proteins?)

- The previous studies such as Morishita et al. (J. Pestic. Sci., 39, 76-84, 2014) demonstrate the Kd of PonA and IC50 of tebfenozide for various insect EcR/USP complex. Please discuss on the difference between P. xylostella and the others. And Morishita et al. have performed in silico docking study and described the important roles of the representative amino acid residues such as Asn503 on the interaction between ligands and receptors. The authors should discuss them in detail.

- L347 deuterated -> tritiated

Author Response

Feng et al. describe that the interaction between the insect ecdysone receptor and Ponasterone A. There are many unclear parts, and the authors should revise the manuscript.

Response: We sincerely thank you for your valuable feedback. According the comments, we have modified our manuscript. We have invited native English scientist from AJE to edit our revised manuscript again. All changes of our manuscript were labelled with blue-colored text. The point to point responds to the comments are listed as following.

- Introduction seems too long. Please make it smart. For example, the topics related to the medical fields in LL31-49 and in LL 56-59 are irrelevant to the topics of this manuscript. And the description in LL 63-64 are also unnecessary.

Response: Thank you for your instructive suggestions. According to your suggestions, we have deleted the parts that are irrelevant to the manuscript and revised the introduction. Please see lines 31-62 in the revised manuscript.

- L73, Since the structure of PoA is strictly different from that of ecdysone or 20-hydroxy ecdysone, please rewrite this part. “Ecdysteroids” are different from “ecdysones”, and PonA is the analogue of ecdysone.

Response: We are grateful for the suggestion. The “Ecdysone (PonA)” has been changed to “The ecdysone analog PonA”. Please see line 50 in the revised manuscript.

- L79 “dihydrazide” -> “dibenzoylhydrazine” ?

Response: Thanks for your careful checks. “dihydrazide” has been corrected as “dibenzoylhydrazine”. Please see line 57 in the revised manuscript.

- L81 The meaning of the description “are difficult to isolate” is unclear.

Response: Thanks for your question. We are sorry this part is not clear. The sentence was modified as “Ecdysone agonists, including commercialized dibenzoylhydrazine insecticides, have been developed to mimic the activity of ecdysteroids and interact with ecdysone receptor proteins, resulting in the continuous excessive molting and death of insects.” Please see lines 57-59 in the revised manuscript.

- L126 “Escherichia. coli” -> “Escherichia coli” (italic and remove a period after Escherichia.

Response: Thank you for your careful reading of our manuscript. We have corrected it. Please see line 103 in the revised manuscript.

- L128 “Komas” -> “Coomassie”?

Response: We sincerely thank you for suggestion. “Komas” has been revised to “Coomassie”. Please see line 105 in the revised manuscript.

- L132 and Figure 1B How did the authors prepare the specific GST-tagged antibody? The Western blotting experiment demonstrates many bands other than the target proteins, even though the specific antibodies were employed. What are these bands? Did the authors further purify the target proteins by using a gel filtration chromatography before use to remove these unnecessary proteins? And can the authors confirm the presence of the heterodimer by the gel filtration chromatography? (the molecular weight of the EcR/USP dimer is 72+68=140 kDa. The retention time of the dimer peak is expected to be clearly different from that of each monomer peak.)

Response: Thanks for the questions. We used a commercial GST-labeled antibody. Many bands appear in the Western blotting due to slight degradation of the protein during long-term storage and contamination that we may have accidentally introduced during the preparation of the protein samples. So we used the fresh target proteins and did this experiment again, as shown below Figure 1B. We used this new picture as Figure 1B in our revised manuscript.

And the bands above PxGST-USP shown in Western blotting experiment may be caused by post-translational modification, while the bands below the target proteins may be caused by slight degradation. However, due to sensitivity of Western blotting, the bands are relatively bright. In addition, we did not use gel filtration chromatography to further purify the target proteins, as a radioisotope assay was used to test protein-ligand activity. For the radioisotope assay, PonA did not bind to the monomeric protein PxGST-USP, but bound specifically to PxGST-EcR; in addition, the binding of PxGST-EcR to [3H]PonA was significantly enhanced in the presence of PxGST-USP (PxGST-EcR and PxGST-EcR/USP dimer-specific binding of 345 dpm and 1179 dpm, respectively) by approximately 4-fold, which was consistent with previous tests. TEM observation, zeta potential analysis and radioisotope experiments in the manuscript could directly and indirectly prove the presence of EcR/USP dimers.

- Figure 2 I cannot believe that TEM can distinguish between a monomer (70 kDa) and a dimer (140 kDa), although I have ever employed TEM before. Please cite some references, in which the sizes of proteins are distinguished using the normal TEM technologies.

Response: Thank you for your valuable and thoughtful comments. Firstly, morphologies of the TEM images of three samples were different in Figure 2. We randomly selected 30 particles in each of the three samples to measure approximately the size by eyes. Sizes change of samples indicated that polymerization occurred in the samples after incubation. Secondly, Isabelle, M. L. B. et al. have proved that the polymerization of EcR and USP of Heliothis virescens exists in the form of dimers by electrospray ionization mass spectrometry and X-ray diffraction (Nature. 2003, 426, 91-96). So we inferred that dimer formation.

- L179-197, and Table 1 Please add the descriptions about the statistical analyses. Have the authors analyzed the significant difference among the data by one-way ANOVA/t-test? If yes, please add the asterisks, *, to display the significant difference (especially for L186, “a significant negative shift”). In LL192-194, the authors write that the overall solution potential was shifted positively when PonA was added to the heterodimer solution. Does this description indicate the change of “-8.56 to -8.23”? If yes, the difference between these values seems insignificant because the standard deviation of -8.56, 1.7, is too big (overlap with -8.23). Related to these, the date of “PxGST-EcR-PonA” and “PxGST-USP-PonA” (and PonA only) should be necessary to compare the data. In addition, for which was the concentration in Table 1, total proteins or each protein? If for each protein, the authors should measure the zeta values for the monomer proteins at 0.5 ug/mL. Please write the concentration of the ligand PonA employed? Please describe the influence of the 10 min sonication (L341) on the proteins. (Continuously sonicated? If yes, the solution will become hot, denaturing proteins?)

Response: Thank you for your careful reading of our manuscript. We have used one-way ANOVA to analyze the significant differences between the data and annotated them in Table 1. In LL192-194, our description shows a slight positive shift of the overall solution potential from -8.56 to -8.23 when PonA is added to the heterodimer solution, comparing only the potential change before and after dimer binding with PonA. Whereas the description in LL162-165 is that the potential change between monomer and dimer solutions in the absence of PonA, which demonstrates dimer formation and therefore does not include the difference comparison between monomer and dimer binding with

PonA. More detailed experiments on the monomer and dimer binding activity of PonA are described in section 2.4 of the revised manuscript. In addition, the concentration in Table1 is total proteins (the total concentration in Table 1 has been corrected as 0.50ug/mL). The concentration of the ligand PonA used is 10 nmol·L-1, and sonication on the proteins is described below “…sonicated for 10 min in the ice-water mixture (ultrasound for 3s, off for 6s, pause for 30s).” Please see lines 171, 175-177 and 327 in the revised manuscript.

- The previous studies such as Morishita et al. (J. Pestic. Sci., 39, 76-84, 2014) demonstrate the Kd of PonA and IC50 of tebfenozide for various insect EcR/USP complex. Please discuss on the difference between P. xylostella and the others. And Morishita et al. have performed in silico docking study and described the important roles of the representative amino acid residues such as Asn503 on the interaction between ligands and receptors. The authors should discuss them in detail.

Response: Thank you for your instructive suggestions. We have added these two parts in the section of discussion. “In addition to P. xylostella (L. ), Morishita, C. M. et al. reported that PonA could bind to the EcR/USP complex from Harmonia axyridis and Epilachna vigintioctopunctata, but tebufe-nozide was inactive against E. vigintioctopunctata [32]. These experiments further showed that the binding activity of PonA to EcR/USP complexes of different species was greater than that of tebfenozide, which bound only partially to EcR/USP complexes in lepidop-teran and coleopteran insects.” “Asparagine plays an important role in the binding of steroid and nonsteroid ligands to EcR of different species. For example, Tribolium confusum TcEcR Asn521 [37], Bemisia tabaci BtEcR Asn309 [38], Heliothis virescens HvEcR Asn504 [39] [29] [32], and Harmonia axyridis HaEcR Asn207 [32].” Please see lines 205-210 and 268-271 in the revised manuscript.

- L347 deuterated -> tritiated

Response: Thank you for your reminder and we have corrected “deuterated” as “tritiated”. Please see line 333 in the revised manuscript.

Thank you for your careful review. We really appreciate your efforts in reviewing our manuscript during this unprecedented and challenging time. We wish good health to you, and your family. Your careful review has helped to make our study clearer and more comprehensive.

Reviewer 2 Report

In this paper, EcR/USP of Plutella xylostella were obtained, the physical size, zeta potential, and molecular weight of the heterodimer were characterized as well. The affinity of the heterodimeric protein to PonA and tebufenozide was determined and significantly enhanced compared with the results of previous in vitro transcriptional translation protein tests and other insect-derived yeast and insect cell line transcriptional activation assays, indicating the high sensitivity of the target protein applied to the radioligand method in this study. Furthermore, this work provides a rapid and sensitive method for screening ecdysone agonists ecdysone agonists for ecdysone receptor studies at the in vitro level. In general, the article is well organized and the data are detailed. I suggest this manuscript can be accepted after minor revision. The comments are as follows:

1.      In section 2.6, hydrogen bond interaction was important, so the distance of hydrogen bond should be calculated and added.

2.      In section 3, some units failed to unified, e.g. mM or mmol·L-1.

3.      In section 3.2, 3.3 and 3.5, some related references need be added.

4.      Some proper nouns should be added the full name when it first appearance, e.g. EcR/USP.

Author Response

In this paper, EcR/USP of Plutella xylostella were obtained, the physical size, zeta potential, and molecular weight of the heterodimer were characterized as well. The affinity of the heterodimeric protein to PonA and tebufenozide was determined and significantly enhanced compared with the results of previous in vitro transcriptional translation protein tests and other insect-derived yeast and insect cell line transcriptional activation assays, indicating the high sensitivity of the target protein applied to the radioligand method in this study. Furthermore, this work provides a rapid and sensitive method for screening ecdysone agonists ecdysone agonists for ecdysone receptor studies at the in vitro level. In general, the article is well organized and the data are detailed. I suggest this manuscript can be accepted after minor revision. The comments are as follows:

Response: We sincerely thank you for your valuable feedback. According the comments, we have modified our manuscript. We have invited native English scientist from AJE to edit our revised manuscript again. All changes of our manuscript were labelled with blue-colored text. The point to point responds to the comments are listed as following.

  1. In section 2.6, hydrogen bond interaction was important, so the distance of hydrogen bond should be calculated and added.

Response: We are grateful for your advice, and the distance of hydrogen bond have been calculated and added in section 2.6. Please see line 257、Figure 4C and Figure 4D in the revised manuscript.

  1. In section 3, some units failed to unified, e.g. mM or mmol·L-1.

Response: Thank you for your careful review. We have unified the unit of this section as “ mmol·L-1”. Please see lines 333-335 in the revised manuscript.

  1. In section 3.2, 3.3 and 3.5, some related references need be added.

Response: We appreciate your advice, References 40, 41, and 44 have been added to sections 3.2, 3.3 and 3.5 of the revised manuscript. Please see lines 287, 308 and 328 in the revised manuscript.

  1. Some proper nouns should be added the full name when it first appearance, e.g. EcR/USP.

Response: Thank you for your suggestion. We are very sorry for our miswriting and they have been corrected. Please see line 9 in the revised manuscript.

Thank you for your careful review. We really appreciate your efforts in reviewing our manuscript during this unprecedented and challenging time. We wish good health to you, and your family. Your careful review has helped to make our study clearer and more comprehensive.

Reviewer 3 Report

In this manuscript, Feng et al. report the expression EcR and USP functional domains of the diamondback moth, Plutella xylostella, using a prokaryotic expression system. The recombinant EcR and USP were purified. Their monomeric and dimeric properties were characterized using transmission electron microscopy and zeta potential. Also, they investigated their binding to PonA and tebufenozide by the radioisotope experiments and theoretically molecular docking approach. There are several major issues resulting in my concern about the results. These issues must be well dealt with before acceptance.

Major issues:

1.     The manuscript is not well written. The text must be edited in detail for improving the witting before acceptance. I have listed some suggestion about how to improve the text in the abstract sect in the minor issues.

2.     The title of section 2.1 includes the identification of target proteins investigated in the study, but there are no any results about this in this section.

3.     According to figure 1, the purified recombinant proteins are not in high purity products. Thus, the authors should explain the results of binding of PonA to EcR/USP are confident. The contaminant proteins will influence the binging between them and the binding of tebufenozide to EcR/USP.

4.     GST tag is not excised from the recombinant proteins. As the molecular weight of GST tag is relatively large, it will influence the structure of the recombinant proteins. Why the authors do not excise the GST tag before using the recombinant proteins to conduct the binding experiments.

Minor issues:

1.     Line 9-12: The sentence is so long. Thus, I can’t quickly go through it. Please try to use much more concise sentences to write the manuscript.

2.     Line 15: “while gradually developing”— “while it or this pest has developed”.

3.     Line 21, delete “And”.

4.     Line 24: delete “also”.

5.     Line 25: delete “ecdysone agonists”.

6.     Line 26: “ecdysone agonists”--“ecdysone agonist”; “binding effects”--“binding effect”.

7.     Line 24-25: This sentence is very confused. Please rewrite it. Besides abstract, there are many typos and confused sentences in the main text of the manuscript. I just list part of them in the below. The authors should carefully edit the main text.

8.     Line 124: delete “Subsubsection”.

9.     Line 126: “Escherichia. coli”---“Escherichia coli”.

10.  The method for section 3.7 is simple.

11.  The section of “4. Conclusions” should be moved to the section following the “Results and Discussion” section.

Author Response

In this manuscript, Feng et al. report the expression EcR and USP functional domains of the diamondback moth, Plutella xylostella, using a prokaryotic expression system. The recombinant EcR and USP were purified. Their monomeric and dimeric properties were characterized using transmission electron microscopy and zeta potential. Also, they investigated their binding to PonA and tebufenozide by the radioisotope experiments and theoretically molecular docking approach. There are several major issues resulting in my concern about the results. These issues must be well dealt with before acceptance.

Response: We sincerely thank you for your valuable feedback. According the comments, we have modified our manuscript. We have invited native English scientist from AJE to edit our revised manuscript again. All changes of our manuscript were labelled with blue-colored text. The point to point responds to the comments are listed as following.

Major issues:

  1. The manuscript is not well written. The text must be edited in detail for improving the witting before acceptance. I have listed some suggestion about how to improve the text in the abstract sect in the minor issues.

Response: Thank you for the important advice. The manuscript has been rewritten, with substantial modification, incorporating other reviewer’s and your points. In order to improve the quality of  our manuscript, we have invited native English scientist from AJE to edit our revised manuscript.

  1. The title of section 2.1 includes the identification of target proteins investigated in the study, but there are no any results about this in this section.

Response: Thank you for your instructive suggestion. We have added the description of the results in section 2.1 as follows “…and the molecular weight where bands appeared matches the theoretical molecular weight of the target protein. Therefore, both the molecular weight and specific antibody detection results of the purified proteins indicate that they are the target proteins.” Please see lines 109-112 in the revised manuscript.

  1. According to figure 1, the purified recombinant proteins are not in high purity products. Thus, the authors should explain the results of binding of PonA to EcR/USP are confident. The contaminant proteins will influence the binging between them and the binding of tebufenozide to EcR/USP.

Response: Thank you for your instructive suggestions. The purity of protein could influence the binding experiment with ligands. We had also considered and reduced the effect of protein purity. The previous study from Baozhen Tang et al. (BMC Mol Biol., 13, 2012) successfully determined the Kd of PonA through in vitro translation of PxEcR and PxUSP, and the target proteins they used was much lower in purity than our purified recombinant proteins. Moreover, our previous study also determined the IC50 of ligands with in vitro-translated Plutella xylostella ecdysone receptor (PxEcR and PxUSP) with the radioisotope assay (Chem Biol Drug Des., 97, 184-195, 2021). In our present study, we obtained EcR and USP with higher purity than Tang, B. Z.’s and our previous EcR and USP. 

  1. GST tag is not excised from the recombinant proteins. As the molecular weight of GST tag is relatively large, it will influence the structure of the recombinant proteins. Why the authors do not excise the GST tag before using the recombinant proteins to conduct the binding experiments.

Response: We are grateful for the suggestion. In fact, we have excised GST tag before recombinant proteins, but the target proteins degraded rapidly and subsequent test experiments could not be completed. Then, the GST tag was retained. Moreover, the Kd value of PonA tested by us indicates that the recombinant proteins with GST tag have bioactive. So we selected to keep GST tag with the recombinant proteins.

Minor issues:

  1. Line 9-12: The sentence is so long. Thus, I can’t quickly go through it. Please try to use much more concise sentences to write the manuscript.

Response: Thank you for your precious advice. We have modified this section description as follows “Both insect ecdysone receptors and ultraspiracle belong to the nuclear receptor family. They form a nanoscale self-assembling complex with ecdysteroids in cells, transit into the nucleus, bind with genes to initiate transcription, and perform specific biological functions to regulate the ecdysone, metamorphosis and growth processes of insects.” Please see lines 9-12 in the revised manuscript.

  1. Line 15: “while gradually developing”— “while it or this pest has developed”.

Response: Thanks for your question. This sentence has been rewritten as follows “The diamondback moth (Plutella xylostella) is a devastating pest of cruciferous vegetable crops, wreaking havoc worldwide and causing severe economic losses, and this pest has developed resistance to most chemical insecticides”. Please see lines 13-15 in the revised manuscript.

  1. Line 21, delete “And”.

Response: Thanks for your advice. “And” was deleted. Please see line 21 in the revised manuscript.

  1. Line 24: delete “also”

Response: Thanks for your carefully reading. “also” was deleted in this sentence. Please see line 24 in the revised manuscript.

  1. Line 25: delete “ecysone agonists”.

Response: We agree with your comment. Repeated “ecdysone agonists” was deleted in this sentence. Please see lines 24-25 in the revised manuscript.

  1. Line 26: “ecdysone agonists”--“ecdysone agonist”; “binding effects”--“binding effect”.

Response: Thanks for your careful checks. We have corrected “ecdysone agonists” as “ecdysone agonist”, “binding effects” as “binding effect”. Please see line 26 in the revised manuscript.

  1. Line 24-25: This sentence is very confused. Please rewrite it. Besides abstract, there are many typos and confused sentences in the main text of the manuscript. I just list part of them in the below. The authors should carefully edit the main text.

Response: Thank you for your careful review. The sentence is described below “This study provides a rapid and sensitive method for screening ecdysone agonists for ecdysone receptor studies in vitro.” in the revised manuscript. Please see lines 24-25. And the manuscript has been thoroughly revised and edited by a native English scientist.

  1. Line 124: delete “Subsubsection”.

Response: Thanks for the suggestion. Subsubsection” was deleted in this sentence. Please see line 101 in the revised manuscript.

  1. Line 126: “Escherichia. coli”---“Escherichia coli”.

Response: Thank you for your careful reading of our manuscript. We are very sorry for our miswriting and have corrected it in the new manuscript. Please see line 103 in the revised manuscript.

  1. The method for section 3.7 is simple.

Response: We have rewritten and add more detail in the methods. “ Homology modeling and molecular docking were performed using MOE 2022.02..….After homology modeling, PxEcR-LBD and PxEcR-LBD were bound to tebufenozide and PonA during molecular docking, respectively. The final docking conformation was selected by the best docking scoring with Generalized Born/Volume Integral (GB/VI) methodology……. The top 20 docked conformations were ranked and clustered. The docking conformations of tebufenozide in Cluster 1 and PonA in Cluster 2 were determined as the main conformations, and their interactions were analyzed and discussed further.” Please see lines 354, 360-363, 366-369 in the revised manuscript.

  1. The section of “4. Conclusions” should be moved to the section following the “Results and Discussion” section.

Response: Thanks for your suggestion. We prepared this manuscript according to the template of molecules. “Materials and Methods” follows “Results and Discussion”. Thank you again.

Thank you for your careful review. We really appreciate your efforts in reviewing our manuscript during this unprecedented and challenging time. We wish good health to you, and your family. Your careful review has helped to make our study clearer and more comprehensive.

Round 2

Reviewer 1 Report

The authors properly take my comments on their original manuscript into consideration in their revision process. I have only one suggestion as described below.  

L12  ecdysone -> molting, ecdysis (please check the meanings of ecdysis and molting)

Author Response

Comments and Suggestions for Authors

The authors properly take my comments on their original manuscript into consideration in their revision process. I have only one suggestion as described below.  

Response: We sincerely thank you for your valuable suggestion and we have made modification to our manuscript. In this revised version, changes to our manuscript were all highlighted within the document using blue-colored text. The response to the comment is listed as following:

L12  ecdysone -> molting, ecdysis (please check the meanings of ecdysis and molting)

Response: We are grateful for your suggestion. After carefully check, the “ecdysone” should have been changed to “molting”. Please see line 12 in the revised manuscript.

Reviewer 3 Report

All comments are well responed. The changes have improved the manuscript.

Author Response

Comments and Suggestions for Authors

All comments are well responed. The changes have improved the manuscript.

Response: We appreciate your enthusiastic work to help us improve the manuscript.